# RNA-Binding Protein-Mediated Alternative Splicing Regulates Abiotic Stress Responses in Plants

**DOI:** 10.3390/ijms251910548

**Published:** 2024-09-30

**Authors:** Ying Guo, Xudong Shang, Ligeng Ma, Ying Cao

**Affiliations:** College of Life Sciences, Capital Normal University, Beijing 100048, China; 2200801012@cnu.edu.cn (Y.G.); 2160801011@cnu.edu.cn (X.S.); ligeng.ma@cnu.edu.cn (L.M.)

**Keywords:** alternative splicing, RNA-binding protein, spliceosome, splicing sites, abiotic stress, plant

## Abstract

The alternative splicing of pre-mRNA generates distinct mRNA variants from a pre-mRNA, thereby modulating a gene’s function. The splicing of pre-mRNA depends on splice sites and regulatory elements in pre-mRNA, as well as the snRNA and proteins that recognize these sequences. Among these, RNA-binding proteins (RBPs) are the primary regulators of pre-mRNA splicing and play a critical role in the regulation of alternative splicing by recognizing the elements in pre-mRNA. However, little is known about the function of RBPs in stress response in plants. Here, we summarized the RBPs involved in the alternative splicing of pre-mRNA and their recognizing elements in pre-mRNA, and the recent advance in the role of RBP-mediated alternative splicing in response to abiotic stresses in plants. This review proposes that the regulation of pre-mRNA alternative splicing by RBPs is an important way for plants to adapt to abiotic stresses, and the regulation of alternative splicing by RBPs is a promising direction for crop breeding.

## 1. Introduction

As sessile organisms, plants are constantly challenged by various stresses that affect their growth and development. The alternative splicing of pre-mRNA is recognized as an important regulatory mechanism in plants to respond to both endogenous and exogenous stimuli [1,2,3]. Pre-mRNA splicing is efficiently and accurately catalyzed by the spliceosome. The spliceosome is a megadalton complex that contains five small ribonucleoproteins (U1, U2, U4, U5, and U6 snRNPs) and more than one hundred proteins. The correct recognition of the splicing sites is the first step in pre-mRNA splicing. U1 snRNP, SF1 and U2AF65 recognize the 5′SS, branch site, and 3′SS, respectively, in initial assembly of the spliceosome [4,5,6]. The active spliceosome is assembled on the pre-RNA and disassembles after completing the intron excision and exon ligation [7,8].

The recognition of different splicing sites by the spliceosome leads to the alternative splicing of the pre-mRNA. The alternative splicing of the pre-mRNA allows a single pre-mRNA to generate different variants, thereby increasing the complexity of the transcriptome and proteome. In rice and maize, at least 50% of the pre-mRNAs undergo alternative splicing [9,10], while approximately 60–80% pre-mRNAs undergo alternative splicing in *Arabidopsis* [11]. There are at least four types of alternative splicing according to the difference in splicing sites used in different introns; they are exon skipping (ES), alternative 5′ splice site (A5SS), alternative 3′ splice site (A3SS), and intron retention (IR) [8]. High-throughput sequencing results show that intron retention is a common type of alternative splicing in plant cells, especially under abiotic stresses such as cold stress or heat shock conditions [12,13,14,15,16]. Intron-retained transcripts are, in most cases, retained in the nucleus, thereby reducing the level of functional splicing variants in the cytoplasm and regulating their protein function [16,17]. However, upon changes in the external environments, these intron-retained transcripts can be further processed by the spliceosome and subsequently transported from the nucleus to the cytoplasm for translation [16,18]. It is hypothesized that the de-repression response generated by the re-splicing of transcripts with retained introns is more advantageous for rapid signal transmission compared to the de novo synthesis of pre-mRNA during transcriptional regulation. Therefore, alternative splicing is a dynamic and efficient regulatory mechanism for plant environmental adaptation.

The dysregulation of alternative splicing in plants affects the adaptability of plants to the changing environment [12,13,14]. Therefore, the precise regulation of alternative splicing is the key for plants to respond to abiotic stresses. RNA-binding proteins (RBPs) are generally regarded as one of the important factors that regulate alternative splicing. During the assembly stage of the spliceosome, alternative splicing is widely regulated by the cis-elements in pre-mRNA and RBPs included in the spliceosome [19,20]. Cis-elements are regulatory elements distributed in pre-mRNA exons and introns that enhance or repress splicing, while RBPs are proteins that recognize those cis-elements. In animal cells, the RBPs that recognize splicing regulatory elements are serine/arginine-rich (SR) proteins and heterogeneous nuclear ribonucleoproteins (hnRNPs). SR proteins can recognize and bind exon splicing enhancers (ESEs) and interact with U2AF and U1 snRNP, thereby promoting the assembly of early splicing complexes at the splicing sites. hnRNP family proteins usually bind exon splicing silencers (ESSs) and play an opposite role to SR proteins, to inhibit the assembly of splicing complexes [21,22,23,24].

During evolution, the proteins required for post-transcriptional gene expression regulation are conserved across eukaryotic lineages [25]. RBPs have also been shown to have conserved functions in regulating alternative splicing due to their conserved RNA-binding domains [26]. Numerous studies have reported that RBPs regulate plant abiotic stress responses [27,28]. In this review, we summarized the recent advance in the function of RBPs in the alternative splicing of pre-mRNAs and in responses to abiotic stresses in plants.

## 2. Pre-mRNA Splicing and Spliceosome Cycle

Most of the genes in eukaryotes are split genes, which are interrupted by multiple exons and introns. The production of mature mRNA requires the splicing of introns in pre-mRNA and the ligation of adjacent exons. The defect in pre-mRNA splicing often leads to the abnormality of gene expression regulation and affects the growth, development, and environmental adaptability of eukaryotes. To ensure the accuracy of gene expression and protein translation, the spliceosome is required to achieve single-base accuracy in the recognition and catalysis of pre-mRNA introns.

The accurate splicing achieved by the spliceosome strictly depends on the sequence-conserved splicing signals in introns. The splicing signals mainly include the 5′ splicing site (5′SS), branch point (BP), and 3′ splicing site (3′SS). The sequence of the 5′SS is GU in most cases. Its downstream element sequence (base at positions +3 to +6) is conserved as AAGU or AUGU in yeast, while it is RAGN with less conservation in vertebrates [29,30]; the 3′SS is a sequence-conserved AG, and the intronic region upstream of it often contains the polypyrimidine tract (PPT). By analyzing the PPT of about 1.2 million introns from 22 eukaryotes, it is found that the PPT signal is weak in most of fungi, intermediate in plants and protozoa, and strong in metazoans, and the strength of the PPT signal is correlated to the key amino acid sequence of splicing factor U2AF2 [31]; BP is a conserved adenylate located 18–40 nucleotides upstream of 3′SS. In *Saccharomyces cerevisiae*, BP is located at the sixth position of the heptamer UACUAAC, but this signal is weakened in vertebrates [30].

The spliceosome is currently the largest known complex in eukaryotes, which contains five small ribonucleoproteins (U1 snRNP, U2 snRNP, U4 snRNP, U5 snRNP, and U6 snRNP) and more than one hundred proteins [7,32]. Pre-mRNA splicing by the spliceosome mainly undergoes four stages—assembly, activation, catalysis, and disassembly [7,32]. To ensure that the catalytic reaction can be carried out accurately and efficiently, the binding and dissociation of splicing factors and snRNA continuously occur on the splicing site of pre-mRNA.

During pre-mRNA splicing, the dynamics of snRNA and splicing factors in the spliceosome are as follows: the first step of pre-mRNA splicing is the base pairing of U1 snRNA with the 5′SS within pre-mRNA intron; the splicing factor 1 (SF1)/branch point binding protein (BBP) binds to the BP sequence of pre-mRNA, and the U2AF65-U2AF35 heterodimer cooperatively binds to the polypyrimidine tract and 3′SS to form the E complex [4]. Under the assistance of U2AF, U2 snRNA replaces SF1 and binds to the BP by means of base pairing to form the A complex [33,34]. At this time, U1 and U2 recruit U4/U6.U5 tri-snRNP to form the pre-B complex, thus completing the assembly of the spliceosome [34,35,36,37]; secondly, the molecular rearrangement of snRNAs and proteins within the pre-B complex forms the B complex. Subsequently, with the dissociation of U1 snRNP from the B complex, the 5′ exon and 5′SS are recognized by loop I of U5 snRNA and the ACAGA box of U6 snRNA, respectively. Then, the dissociation of U4 snRNP and the recruitment of NTC/NTR to the B complex promote the formation of the splicing active site of RNA in the B^act^ complex [38]. Subsequently, through the dissociation of some splicing factors, the BP is delivered to the 5′SS/U6 duplex in the splicing active site, thus completing the activation of the spliceosome (B* complex) [38,39,40,41]; thirdly, the two-step transesterification reaction is catalyzed by the spliceosome. For the branching reaction, the BP adenosine nucleophile attacks the phosphorus of the 5′SS to result in the liberation of the 5′ exon and the formation of an intron lariat–3′ exon intermediate, thereby forming the C complex [42]. Then, the 3′SS-3′ exon replaces the 5′SS/U6 snRNA duplex and enters the splicing active center in the C* complex. In the second phosphoryl transfer reaction, the 3′OH group of 5′ exon attacks the phosphorus atom of the 3′SS, and the exons are ligated in the P complex (post-catalytic spliceosome) and are released from the spliceosome (intron lariat spliceosome, ILS) [43,44]; and fourthly, the intron with the lariat structure is released from the spliceosome and rapidly degraded, and the spliceosome disassembly and enters the next cycle [7,32].

## 3. The Spliceosomal RBPs and Their RNA-Binding Elements in Plants

A pre-mRNA has multiple exons and introns. Even an intron often contains multiple splicing sites, but not every splicing site is used during splicing. The selection of splicing sites is the prerequisite for the initial assembly of the spliceosome. Competition between different splicing sites will lead to alternative splicing, thus enriching the transcriptome and proteome of eukaryotes [45]. In eukaryotes, splicing regulatory cis-elements (SREs) and trans-acting factors (RNA-binding proteins) jointly regulate the selection of splicing sites.

### 3.1. Splicing Regulatory Cis-Elements and Trans-Acting Factors in Splicing Regulation

On a transcriptome-wide scale, there is only half of the information for pre-mRNA splicing provided by splicing sites. The remaining information is mainly provided by SREs around the splicing sites [46,47]. SREs are splicing enhancers or splicing silencers distributed in pre-mRNA exons and introns (ESEs, ISEs, ESSs, or ISSs). Hundreds of splicing regulatory elements distributed in human cells have been discovered using genome-wide analysis [48,49,50]. The splicing regulatory elements often have four characteristics: 1, additive effect. The functions of splicing enhancers or silencers are usually additive; 2, synergistic effect. Different SREs may also synergistically regulate alternative splicing. For example, the exon UAGG element that overlaps with 5′SS and the intron GGGG element can act synergistically; 3, context-dependent activity. The same sequence has different functions for splicing when it is located in different genes or exons/introns of the same gene. For example, placing ISSs in exon positions can activate or inhibit exon inclusion and inhibit or enhance the use of proximal 5′SS for different pre-mRNA [50]. In addition, similar elements have different regulatory effects on splicing at different positions in the same intron. For example, elements consisting of CA repeats or CA-rich elements can act as ISEs or ISSs depending on their distance from the upstream exon [51]; 4, splicing regulatory network formation. Each RNA element is recognized by multiple factors, and most factors bind multiple SREs’ elements, thus forming a splicing regulatory network [50].

The RBPs that recognize and bind splicing regulatory elements mainly include two families, SR proteins and hnRNPs. In animal cells, SR proteins can mainly recognize and bind exon enhancer element ESEs and mediate protein interactions; the SR proteins interact with U2AF and U1 snRNP, thereby promoting the assembly of early splicing complexes at splicing sites. In some cases, the SR proteins can interact with the BPS and 5′SS of pre-mRNA during the assembly of the splicing complex [52,53]; hnRNPs are a group of proteins containing one or more RBDs (mainly RRM), and sometimes contain splicing inhibitory domains, such as glycine-rich motifs [54]. hnRNPs usually bind to ESSs or ISSs and have the opposite effect to SR proteins, that is, inhibiting the recognition of splicing sites and inhibiting the assembly of splicing complexes on pre-mRNA [21,22,23,24]. For example, PTB (hnRNP I) can block the basic interaction between U1 snRNP and U2 snRNP [55,56], and hnRNP A1 can inhibit splicing by binding on both sides of an exon and “looping out” the exon or directly replacing snRNP binding [57].

### 3.2. The Spliceosomal RBPs in Plants

Splicing factors are conserved in eukaryotes [25]. Koncz et al. found that there are approximately 430 splicing factors in *Arabidopsis* [58]. Among them, 43 RBPs, including 15 SR proteins and SR-related proteins and 28 hnRNP-like proteins in *Arabidopsis*, are homologous of SR and hnRNP proteins in human (Table 1). These splicing factors contain conserved RNA-binding domains (RBDs), which include the RNA recognition motif (RRM), hnRNP K homology (KH), and zinc finger (Znf) (Table 1).

The recognition and binding of splicing regulatory elements by RBPs is generally achieved through the binding between the RBDs of RBPs and RNA elements in pre-mRNA. RBDs are conserved domains from bacteria to animals and plants [59,60]. There are numerous types of RBDs, and they are usually short in length, typically around 80 amino acids, and only a few of their amino acid residues are responsible for its binding to RNA elements [61]. The RRM is a classical RBD and is widely presented in plant RNA processing factors [27]. Each of the RNP1 and RNP2 motifs in the RRM contain three conserved amino acid residues—arginine or lysine plus two aromatic amino acid residues, and they are responsible for binding to single-stranded RNA [62,63]; the KH domain of RBP binds single-stranded RNA through its conserved motif VIGXXGXXI [64,65]. The KH domain usually appears in tandem (such as the homologous protein PEP of hnRNP F) or appears in combination with other RRMs (such as UBA2a) in plants (Table 1); the ZnF motif of RBPs was originally discovered in transcription factors, and its cysteine and histidine residues are responsible for binding RNA under the coordination with zinc atoms [66,67]. The zinc finger motif often appears together with the RRM in splicing factors, such as SRSF7 homologous proteins RS2Z32 and RS2Z33 (Table 1); the PWI motif is a new type of RBD originally discovered in the human SR-related protein SRm160; it is named after its unique Pro-Trp-Ile feature, and it has a positively charged nucleic acid binding surface [68]. Until now, two RBPs containing the PWI motif have been identified in *Arabidopsis*; they are RBM25 and SRRM1L [69,70].

In addition to RBD, plant splicing factors often contain a type of intrinsically disordered domain (IDR), such as the arginine/serine repeat (RS repeat), which is rich in SR proteins, auxiliary motifs such as GY/RGG/WW in hnRNP-like proteins, and glycine rich motifs in GRP proteins [71,72,73]. They often act as auxiliary subunits with RBD and are responsible for interactions with other splicing factors. The RS domain of SR45 can independently interact with the U2AF35 protein, thereby integrating SR45 into spliceosomes [74].

**Table 1 ijms-25-10548-t001:** *Arabidopsis* RNA-binding proteins homologous to human spliceosomal splicing factors.

RBPs	RBDs	Genes	Metazoan Homologues	Genes (*H*.*s*.)	Spliceosomal Complexes (*H*.*s*.)	References
SR proteins						
SC35	RRM_1	*AT5G64200*	SRSF2	*NP_003007*	A, B, B*, C	[33,75,76,77]
SR33/SCL33	RRM_1	*AT1G55310*	SRSF10	*NP_006616*	B, B*, C	[75,76]
SCL30a	RRM_1	*AT3G13570*				
SCL30	RRM_1	*AT3G55460*				
SCL28	RRM_1	*AT5G18810*				
SR34	RRM_1	*AT1G02840*	SRSF1/ASF/SF2	*NP_008855*	A, B, B*, C	[33,75,76,77]
SR34b	RRM_1	*AT4G02430*				
SR30	RRM_1	*AT1G09140*				
SR34a	RRM_1	*AT3G49430*				
RSZp22/SRZ22	RRM_1, zf-CCHC	*AT4G31580*	SRSF7	*NP_001026854*	A, B (U1), B*, C	[33,75,76,77,78]
RSZp22a	RRM_1, zf-CCHC	*AT2G24590*				
RSzp21/SRZ21	RRM_1, zf-CCHC	*AT1G23860*				
Tra/SFRS1	RRM_1	*AT1G07350*	TRA2B	*NP_004584*	A, B, B*, C	[33,75,76,77]
-	RRM_1	*AT4G35785*	TRA2A	*NP_037425*	A, B (U1), C	[33,77,78]
SR-related proteins						
SRRM1L	PWI	*AT2G29210*	SRRM1	*NP_005830*	A, B (U1), B*, C	[33,75,76,77,78]
hnRNP family						
RNPA/B_1	RRM_1	*AT4G14300*	HNRNPA3	*NP_919223*	A, B, C	[33,75,76,77]
RNPA/B_2	RRM_1	*AT2G33410*				
RNPA/B_3	RRM_1	*AT5G55550*				
RNPA/B_4	RRM_1	*AT4G26650*				
RNPA/B_5	RRM_1	*AT5G47620*				
MSIL4	RRM_1	*AT3G07810*	HNRNPA2B1	*NP_112533*	A, B (U1), C	[33,75,76,77,78]
RNPA/B_7	RRM_1	*AT1G58470*				
RNPA/B_8a	RRM_1	*AT5G40490*				
RBGD3	RRM_1	*AT3G13224*				
RNPA/B_8b	RRM_1	*AT1G17640*				
RNP_N1	RRM_1	*AT3G13224*				
-	RRM_1	*AT5G46840*				
UBA2a	RRM_1, Nup35_RRM	*AT3G56860*				
UBA2b	RRM_1	*AT2G41060*				
UBA2c	RRM_1	*AT3G15010*				
RBGD1	RRM_1	*AT1G17640*	HNRPDL/DAP40a	*NP_112740*	just identified in spliceosome	[79]
PEP/PEPPER	KH_1, KH_2	*AT4G26000*	HNRNPF	*NP_004957*	B (U1), C	[75,76,77,78]
hnRNP-H/RNPH/F_1	RRM_1	*AT5G66010*				
hnRNP-H/RNPH/F_2	RRM_1	*AT3G20890*				
hnRNP-G1	RRM_1, zf-CCHC	*AT5G04280*	RBMX/hnRNP G	*NP_002130*	A, B (U1), B*, C	[33,75,76,77,78]
RZ-1A/hnRNP-G2	RRM_1, zf-CCHC	*AT3G26420*				
RZ-1B/hnRNP-G3	RRM_1, zf-CCHC	*AT1G60650*				
PTB3	RRM_1, RRM_5	*AT1G43190*	PTBP1/hnRNP I	*NP_787041*	just identified in spliceosome	[79]
PTB1	RRM_1, RRM_5	*AT3G01150*				
PTB2	RRM_1, RRM_5	*AT5G53180*				
LIF2/hnRNP-R1	RRM_1	*AT4G00830*	HNRNPR	*NP_005817*	A, B (U1), B*, C	[33,75,76,77,78]
GRP7	RRM_1	*AT2G21660*	CIRBP	*NP_001271*	just identified in spliceosome	[79]
GRP8	RRM_1	*AT4G39260*				

### 3.3. Plant RBP Binding Elements in Pre-mRNA

The recognition and binding of RNA by RBPs is crucial for the processing of pre-mRNA. The studies on the binding of regulatory elements by RBPs initially started from in vitro assays in plants, such as the electrophoretic mobility shift (EMSA) and systematic evolution of ligands by exponential enrichment (SELEX). Subsequently, the transcriptome sequencing analysis of alternative splicing events led to the in vivo investigation of the RNA elements bound by splicing factors. Later, RNA immunoprecipitation sequencing (RIP-seq) and cross-linking immunoprecipitation sequencing (CLIP-seq) were used to describe the binding landscape and explore the binding elements of splicing factors in plants.

Thomas et al. identified the first intron cis-element binding by the SR protein in plants, which is the GAAG repeat element bound by SCL33 [80]. EMSA experiments demonstrated that SCL33 binds to the GAAG element in the third intron of its own pre-mRNA. SCL33 directly interacts with U1-70K to regulate splicing at its 3′ splice site (Table 2). Using a similar approach, the pyrimidine-rich RNA elements bound by PTB to ES were identified in *Arabidopsis* [26]. Currently, only a few splicing factors, including SR45, SRRM1L, RZ-1C, and GRP7, which bind RNA elements in vivo, have been identified [70,72,81,82,83,84].

Overall, it has been demonstrated so far that splicing factors with RNA-binding ability usually bind to degenerate sequences, and most of these proteins contain RRM domains. For example, both the hnRNP-like RBP GRP7 and the GRP protein RZ-1C contain RRM domains and bind to U-rich RNA element (Table 2).

**Table 2 ijms-25-10548-t002:** Splicing-related RBPs and their target elements in *Arabidopsis*.

RBPs	Protein Family	RNA Elements	Target Determination Methods	PPI-Spliceosome Components	AS Events	References
SCL33	SR protein	GAAG repeat	splicing reporter approach; EMSA	U1-70K	A3SS	[80,85]
SC35	SR protein	AGAAGA	matriX motifs (XX motif) method	U1-70K	IR	[85,86]
SR45	SR protein	GGNGG	RIP-seq	U1-70K; U2AF35b	IR; A5SS; A3SS	[74,81,85,87]
		GAA/GA repeat (5′SS) CUU/UC repeat (3′SS)				
SRRM1L	SR-related protein	CU-rich	RIP-seq; RNA-seq; EMSA	U1-70K	IR	[70]
PTB1/2	hnRNP-like	pyrimidine motifs (UC-rich)	splicing reporter analysis; EMSA	U2AF65 (Speculation based on conservative mechanisms)	ES	[26,88]
GRP7	hnRNP-like	GUUUC (U-rich)	iCLIP	U1-70K	A5SS; A3SS; IR	[72,83,84,89]
RZ-1C	glycine-rich protein	U-rich; GA-rich	eCLIP/SELEX	SR proteins	co-transcriptional splicing	[82,90]
GRP20	glycine-rich protein	purine-rich (GA-rich)	mRNA seq; EMSA	PRP18 (U5 snRNP)	ES (for micro- and small exon)	[91]

## 4. RBP-Mediated Alternative Splicing in the Regulation of Plant Abiotic Stresses

When exposed to abiotic stresses, such as drought, high soil salinity, or extreme temperatures, plants reprogram their gene expression in response to the changing environments to enhance their adaptability. Alternative splicing is widely recognized as one of the key regulatory mechanisms to respond to hostile environments in plants. Transcriptomic analyses and genetic studies have shown that RBPs play an important role in plant responses to abiotic stresses by regulating alternative splicing.

### 4.1. Drought Stress Response

Several splicing factors, such as RNA-binding proteins RBM25, SR45, and HIN1, are involved in the regulation of ABA-dependent drought stress responses in plants through mediating the alternative splicing of pre-mRNAs.

RBM25 is a plant RBP with an RRM domain and is also a homologous protein of yeast U1snRNP Snu71 [58]. RBM25 binds to the 5′ splicing site of the last intron of the ABA signal negative regulator *hypersensitive to ABA 1 (HAB1)*pre-mRNA, promotes the recognition of this splicing site and the splicing of the intron, and finally promotes the production of the functional isoform HAB1.1. The phosphatase activity of HAB1.1 inhibits the phosphorylation of SnRK2.6/OST1 to thereby inhibit ABA signal transduction. At the same time, RBM25 inhibits the production of the isoform HAB1.2, which is a truncated protein with the dominant negative effect in *Arabidopsis*, therefore keeping the ABA signaling pathway in an active state [69]. Similarly, RBM25 regulates the alternative splicing of *HAB1* pre-mRNA, leading to a down-regulation of the *HAB1.2*/*HAB1.1* ratio under drought treatment, further regulating drought tolerance in *Arabidopsis* [92]. Furthermore, it has been shown that RBM25 regulates the drought response by regulating ABA-mediated stomatal closure in *Arabidopsis* [92]. Overall, it indicates that RBM25 negatively regulates the ABA-dependent drought stress response by regulating the alternative splicing of *HAB1* pre-mRNA in plants (Figure 1).

RBM25 in U1snRNP can bind to the last intron of *HAB1* pre-mRNA and regulate its alternative splicing, resulting in two splicing variants, *HAB1.1* and *HAB1.2*. *HAB1.1* encodes a full-length protein, and its phosphatase activity inhibits the ABA response, preventing stomata from closing and causing drought intolerance in *Arabidopsis*; *HAB1.2*, which retains introns, encodes a truncated protein with a dominant negative effect. Thus, it can maintain the transduction of the ABA signal, which subsequently leads to stomatal closure and drought tolerance in *Arabidopsis*.

Similar to the function of RBM25, SR45 is involved in the drought stress response by regulating the alternative splicing of pre-mRNA [93]. SR45 is an RBP belonging to the SR protein family. It interacts with spliceosomal components U1-70K and U2AF35b to regulate the selection and the usage of splice sites [74,82,85]. RIP-seq results indicate that SR45 binds to the pre-mRNAs related to ABA signaling and drought stress. Furthermore, SR45 regulates the alternative splicing of *HAB1* pre-mRNA, enhances the function of HAB1, and inhibits ABA signal transduction. Additionally, a drought treatment experiment has shown that SR45 negatively regulates drought tolerance in *Arabidopsis*. These findings suggest that SR45 negatively regulates ABA-dependent drought stress response by modulating the alternative splicing of *HAB1* pre-mRNA [87].

The RNA-binding protein HIN1 is considered a plant-specific RNA-binding protein and is also identified as a splicing factor. HIN1 binds to GAA repeats, which are exon splicing enhancer-like elements, in vitro. Under osmotic stress, HIN1 co-localizes with SR proteins to nuclear speckles, promoting the splicing efficiency of pre-mRNAs of target genes related to response to osmotic stress or signal transduction, resulting in a lower intron-retention rate. These results suggest that HIN1 regulates the alternative splicing of pre-mRNA in response to osmotic stress. Furthermore, *hin1* proteins are sensitive to osmotic stress, and their overexpression lines are resistant to osmotic stress, suggesting that HIN1 proteins promote tolerance to osmotic stress by the regulation of alternative splicing of pre-mRNA in *Arabidopsis* [94].

### 4.2. Salt Stress Response

SCL106 is an RBP associated with the SC subfamily of SR proteins. In rice, a large number of mis-splicing events at the 3′ splicing site were observed in *scl1-6*, indicating that OsSCR106 regulates the selection of the 3′ splicing site in rice. Under salt stress conditions, OsSCR106 regulates the production of splicing variants for multiple pre-mRNAs, indicating that it regulates the alternative splicing events under salt stress. *scr106* exhibits extreme sensitivity to salt stress, further indicating that OsSCR106 increases plant salt tolerance by maintaining the normal splicing of pre-mRNA [95].

The hnRNP-like protein GRP7 is an RBP with an RRM domain in *Arabidopsis*. An iCLIP experiment showed that GRP7 binds to the U-rich RNA element in *Arabidopsis*. GRP7 integrates into the spliceosome mainly by interacting with U1-70K and is responsible for regulating the recognition of 3′ splice sites (Table 2). GRP7 regulates the alternative splicing of many stress-responsive genes, such as ABA signaling and salt stress-related genes [84,89]. Under salt conditions, GRP7 overexpressing lines were more sensitive to salt stress than wild-type plants in *Arabidopsis*, suggesting that GRP7 plays a negative role in salt stress responses in *Arabidopsis* [96]. These results suggest that GRP7 negatively regulates the salt stress response in *Arabidopsis* by mediating the alternative splicing of pre-mRNA.

Recent studies have shown that an RBP SRRM1L is a PWI domain-containing SR-related protein in *Arabidopsis*. It binds to a CU-rich element (Table 2) and regulates the selection of splicing sites by interacting with U1-70K in *Arabidopsis* [70]. Under salt stress conditions, SRRM1L promotes the recognition and splicing of the first intron of *nuclear factor Y subunit A 10* (*NFYA10*), thereby producing a *NFYA10.1* variant, which encodes a functional NFYA10 transcription factor and enhances plant salt tolerance. Mutations in *SRRM1L* lead to the retention of the first intron of *NFYA10*, resulting in the non-functional *NFYA10.3* variant, making plants sensitive to salt stress [70]. The above results indicate that SRRM1L positively regulates salt tolerance by regulating the alternative splicing of *NFYA10* pre-mRNA in plants (Figure 2).

In wild type plants, SRRM1L binds to the 5′ splicing site of the first intron of *NFYA10* pre-mRNA and triggers the splicing mechanism through interaction with U1-snRNP, resulting in the processing of *NFYA10* pre-mRNA into a functional *NFYA10.1* variant, which encodes functional NFYA10 and enhances plant resistance to salt stress. However, in the *srrm1l* plants, the first intron of *NFYA10* pre-mRNA cannot be spliced efficiently. The accumulated abnormal and non-functional *NFYA10.3* variant is directly degraded through the NMD pathway, resulting in plant sensitivity to salt stress.

### 4.3. Heat Shock Response

Some of pre-mRNAs that encode heat stress transcription factors (HSFs), such as *HsfA2* in rice, tomato, and *Arabidopsis*, as well as *HsfA7b*, *HsfB1*, and *HsfB2a* in *Arabidopsis*, are regulated by alternative splicing [97,98,99,100]. The heat shock-induced alternative splicing of the aforementioned pre-mRNAs is often not a process of heat damage, but rather a proactive adaptation to stress by the plants [101]. RNA-binding proteins, such as SR proteins and GRP proteins, regulate the heat stress response by modulating the alternative splicing of *HSF* pre-mRNA in plants [102,103].

SF1 regulates *HsfA2* pre-mRNA alternative splicing by interacting with U2AF65A and U2AF65B, thereby promoting *HsfA2* gene expression in *Arabidopsis* [104]. Lee and colleagues further discovered that AtSF1 regulates the alternative splicing of *HsfA2* pre-mRNA in a manner dependent on its RRM domain, suggesting that the RRM domain of SF1 is crucial for the recognition of *HsfA2* pre-mRNA splicing sites [105]. In addition, *AtSF1* and *AtU2AF65A* mutants showed reduced heat tolerance in *Arabidopsis* [105,106], indicating that SF1 and U2AF65A positively regulate plant heat tolerance. Further study has shown that AtSF1 and AtU2AF65A promote the heat stress response by regulating the alternative splicing of *HsfA2* pre-mRNA, thereby enhancing heat tolerance in *Arabidopsis*. Similar to SF1, SR proteins, such as SR30, SR34a, RS40, and SR45, are all involved in heat stress responses [107].

GRP3 and GRP16 are two glycine-rich RBPs. Under heat shock conditions, the expression of *OsGRP3/OsGRP16* increases at night. In addition, OsGRP3/OsGRP16 interact with U1snRNP and U2snRNP to inhibit exon skipping under heat stress conditions, thereby positively regulating the expression of HSFs and HSPs and modulating diurnal thermotolerance [103].

### 4.4. Cold Stress Response

Alternative splicing is a major mechanism in the regulation of the cold stress response, and rice SR proteins are involved in the plants’ adaptation to cold stress [108]. A recent study shows that two SR proteins, OsRS33 and OsRS2Z38, are up-regulated during cold treatment and induce the alternative splicing of the pre-mRNA of their target genes, indicating that OsRS33 and OsRS2Z38 regulate alternative splicing in response to cold stress in rice [108]. In addition, previous studies found that the *Osrs33* has a cold stress-sensitive phenotype [109], indicating that OsRS33 enhances the tolerance of rice to resist cold stress by regulating pre-mRNA alternative splicing.

As a splicing factor, GRP7 binds to the GUUUC element of pre-mRNAs to regulate their alternative 5′ or alternative 3′ splicing sites in *Arabidopsis* (Table 2). *gpr7* exhibits a cold-sensitive phenotype in *Arabidopsis*, and the overexpression of GRP7 increases the tolerance to cold, indicating that GRP7 enhances cold tolerance by the regulation of the alternative splicing of pre-mRNAs in plants [96,110].

## 5. Conclusions and Prospections

Alternative splicing is an important mechanism for the regulation of abiotic stress responses in plants. In addition to studying the functions of alternative splicing variants of key genes involved in abiotic stress responses, current studies are focusing on how RBPs coordinate the generation of different alternative splicing variants to deal with plant abiotic stresses. In this review, we focus on the vital role of RBP-mediated alternative splicing on drought, salt, heat, and cold stresses (Figure 3). In addition, common abiotic stresses for plants also include waterlogging, freezing, heavy metal ions, and ultraviolet stresses. However, there is very little progress about the participation of RBP-mediated alternative splicing in any of the aforementioned other four abiotic stress responses at present. Therefore, this can be a new direction in the future.

RBPs, such as SR proteins, hnRNP-like proteins, and GRPs, recognize and bind to splicing regulatory elements, and interact with U1 snRNP or U2 snRNP to promote or inhibit the recognition of splicing sites. In most cases, it will lead to a decrease in splicing efficiency, resulting in more transcripts with retained introns under drought, salt, and temperature stresses. These stress-induced alternative splicing variants have several fates—they will remain in the nucleus or be transported out of the nucleus and degraded through the NMD pathway to down-regulate gene expression; in addition, alternative splicing variants exported from the nucleus can also be translated into alternative splicing isoforms that perform synergistic or antagonistic functions (such as dominant negative effects), leading to a special stress-reduced proteome. Thus, RBP-mediated alternative splicing can regulate response to abiotic stress, thereby modulating their adaptability to changing environments in plants (Figure 4).

RBPs specifically recognize and bind to splicing regulatory elements and affect spliceosome assembly by regulating the recognition of splicing sites by U1 snRNP or/and U2 snRNP. Their function on splicing is affected under stress conditions, thereby resulting in more IR variants. Most of them are retained in the nucleus, some are transported to the cytoplasm for translation into stress-induced isoforms, and a few are degraded by the NMD pathway, thus enhancing plant adaptation to the environment.

In addition, the additive effect of splicing regulatory elements will result in enhancing or inhibiting splicing processing [47]. The in tandem increase, mutation, or deletion of splicing regulatory elements in plant introns are some of the ways to regulate pre-mRNA splicing efficiency without affecting the coding sequence. When full-length protein encoding by the target gene is a negative regulator of the stress response, such as in *HAB1* in drought stress (Figure 1 and Figure 3), the deletion or mutation of the splicing enhancer (such as ISE), or tandemly increasing its splicing silencer (such as ISS), will weaken the function of target gene. Conversely, if the full-length protein encoding by the target gene is a positive regulator of the stress response, such as in *NFYA10* in salt stress (Figure 2 and Figure 3), then it is necessary to increase the tandem of ISE or delete ISS to enhance the splicing site, thereby enhancing the function of this protein and promoting plant abiotic stress responses. Therefore, the regulation of splicing by the modulation of splicing regulatory elements in pre-mRNA is an important direction in the field of plant pre-mRNA splicing in the future.

Moreover, abiotic stress seriously affects the yield and quality of crops. Post-transcriptional regulation, especially the regulation of alternative splicing, is generally considered to be a key regulatory process of plants’ response to environmental stress. The regulation of alternative splicing by RBPs could be regarded as a promising direction for crop breeding and improvement under stress conditions. For example, the RBP AtGRP7 interacts with U1-70K to regulate the alternative splicing of pre-mRNA of stress-related genes in *Arabidopsis* (Table 2). A study has shown that transgenic rice lines, which are heterologously expressing the *Arabidopsis* RBPs GRP2 or GRP7, have significantly higher yields compared to non-transgenic rice under drought stress, but without any side-effects on plant size or productivity under normal growth conditions [111], indicating that GRPs can be used to improve the yield potential of crops under stress conditions.

## Figures and Tables

**Figure 1 ijms-25-10548-f001:**
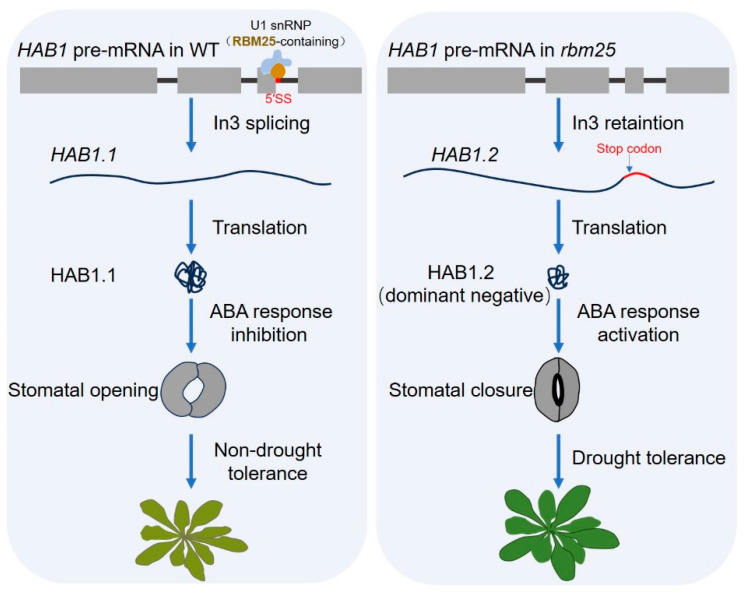
RBM25-mediated *HAB1* pre-mRNA alternative splicing negatively regulates drought tolerance in *Arabidopsis*.

**Figure 2 ijms-25-10548-f002:**
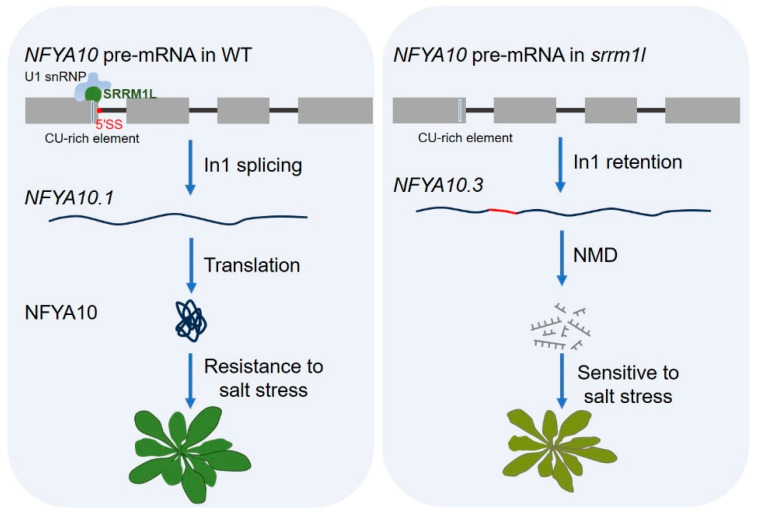
SRRM1L-mediated alternative splicing of *NFYA10* regulates the response to salt stress in *Arabidopsis*.

**Figure 3 ijms-25-10548-f003:**
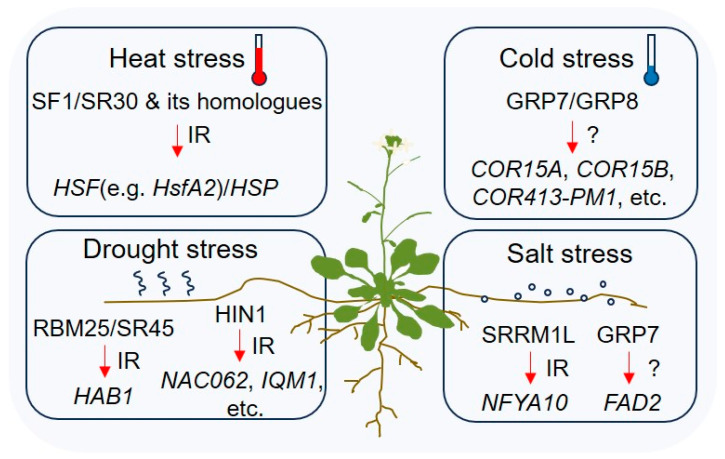
RBPs and their AS-targets involved in abiotic stress in *Arabidopsis*. Red arrow means the regulation of alternative splicing mediated by RBPs.

**Figure 4 ijms-25-10548-f004:**
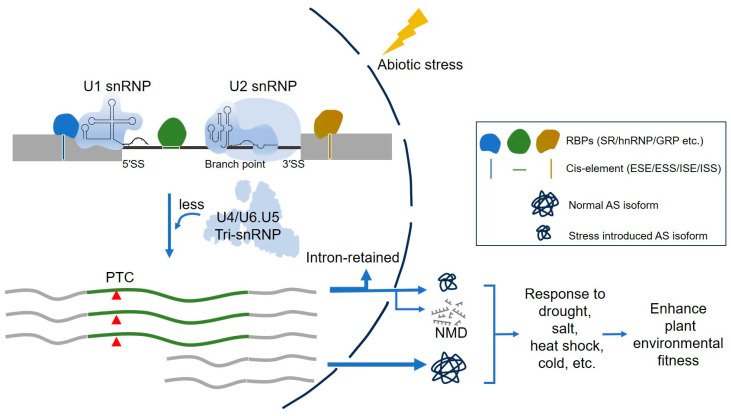
RBP-mediated alternative splicing regulates abiotic stress response in plants.

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
