# Peer review of "RNA-Binding Protein-Mediated Alternative Splicing Regulates Abiotic Stress Responses in Plants"

_ijms, 2024, doi:10.3390/ijms251910548_

Round 1

Reviewer 1 Report

Comments and Suggestions for Authors

The review is great and provides very useful information. Makes a well designed and comprehensive summary of all the information realted to RNA-Binding Protein-Mediated Alternative Splicing to familiarize with the technology to the nonspecialist reader and then in the last part focus on what is known on abiotic stress.

I have only minor advices for improvement:

- The most specific information is found at the end of the review, where it describes what is know for each stress. I would include a figure summarizing the genes regulated by splicing in each stress.

-Authors focus the abiotic stress en salt, drough, heat shock and cold stress. I would also include: waterlogging, freezing, heavy metal and UV stress. Is there anything known under these conditions? Please include this information to complete the review. 

Author Response

The review is great and provides very useful information. Makes a well designed and comprehensive summary of all the information related to RNA-Binding Protein-Mediated Alternative Splicing to familiarize with the technology to the nonspecialist reader and then in the last part focus on what is known on abiotic stress.

I have only minor advices for improvement:

Q1: The most specific information is found at the end of the review, where it describes what is known for each stress. I would include a figure summarizing the genes regulated by splicing in each stress.

Our response: Done as suggested by the reviewer, the genes regulated by splicing in each stress were summarized in a new figure (Figure 3 in revised manuscript).

Q2: Authors focus the abiotic stress en salt, drough, heat shock and cold stress. I would also include: waterlogging, freezing, heavy metal and UV stress. Is there anything known under these conditions? Please include this information to complete the review. 

Our response: There is no report about the involvement of RBP-mediated alternative splicing in waterlogging, freezing, heavy metal or UV stress. We discussed this point and pointed that this is the new direction for the further study in the revised manuscript.

Reviewer 2 Report

Comments and Suggestions for Authors

The review article "RNA-Binding Protein-Mediated Alternative Splicing Regulates Abiotic Stress Responses in Plants" submitted by Ying Cao et al., reports an interesting topic. The authors summarized recent advance in the function of RNA-binding proteins in alternative splicing of pre-mRNAs and in responses to abiotic stresses in plants. This is a well-written review article, and I consider the manuscript suitable for publication subject to the following improvements.

1. Add research gaps in the abstract section, covered in this review.

2. Revise the statement in Line 14–15. "Furthermore, the recent advance in the role of RBP-mediated alternative splicing in response to abiotic stresses in plants was reviewed."

3. Line 52-53: Add relevant reference "The dysregulation of alternative splicing in plants affects the adaptability of plants to the changing environment." 

4. It is suggested to add waterlogging and heavy metal stress responses in Section 4.

5. Figure 3. can you add stress-specific response?

6. Line 378-382 add more details here.

Author Response

The review article "RNA-Binding Protein-Mediated Alternative Splicing Regulates Abiotic Stress Responses in Plants" submitted by Ying Cao et al., reports an interesting topic. The authors summarized recent advance in the function of RNA-binding proteins in alternative splicing of pre-mRNAs and in responses to abiotic stresses in plants. This is a well-written review article, and I consider the manuscript suitable for publication subject to the following improvements.

Q1: Add research gaps in the abstract section, covered in this review.

Our response: Revised as suggested.

Q2: Revise the statement in Line 14–15. "Furthermore, the recent advance in the role of RBP-mediated alternative splicing in response to abiotic stresses in plants was reviewed."

Our response: Revised.

Q3: Line 52-53: Add relevant reference "The dysregulation of alternative splicing in plants affects the adaptability of plants to the changing environment." 

Our response: Relevant references were added in the revised manuscript.

Q4: It is suggested to add waterlogging and heavy metal stress responses in Section 4.

Our response: There is no report about the involvement of RBP-mediated alternative splicing in waterlogging, heavy metal or UV stress. We discussed this point and pointed that this is the new direction for the further study in the revised manuscript.

Q5: Figure 3. can you add stress-specific response?

 Our response: Revised the Figure (Now Figure 4 in the revised manuscript) as suggested

Q6: Line 378-382 add more details here.

Our response: More details were added as suggested.